# Longitudinal assessment of COVID-19 vaccine uptake: A two-wave survey of a nationally representative U.S. sample

Caroline Katzman[1,2], Tucker Morgan[3], Ariel de Roche[1], Julen Harris[1,2], Christine Mauro[3], Gregory Zimet[4], Susan Rosenthal[1,5]*

1 Department of Pediatrics, Vagelos College of Physicians and Surgeons, Columbia University Irving Medical Center, New York, NY, United States of America, 2 NewYork-Presbyterian Hospital, New York, NY, United States of America, 3 Department of Biostatistics at the Mailman School of Public Health at Columbia University Irving Medical Center, New York, NY, United States of America, 4 Division of Adolescent Medicine, Indiana University School of Medicine, Indianapolis, IN, United States of America, 5 Department of Psychiatry at Vagelos College of Physicians and Surgeons at Columbia University Irving Medical Center, New York, NY, United States of America

* slr2154@cumc.columbia.edu

**Data Availability Statement:** Data uploaded to ICPSR. The URL to access the data is as follows: https://www.openicpsr.org/openicpsr/project/193927/version/V1/view

## Abstract

Understanding factors that influence those who are initially COVID-19 vaccine hesitant to accept vaccination is valuable for the development of vaccine promotion strategies. Using Ipsos KnowledgePanel®, we conducted a national survey of adults aged 18 and older in the United States. We created a questionnaire to examine factors associated with COVID-19 vaccine uptake over a longitudinal period ("Wave 1" in April 2021 and "Wave 2" in February 2022), and utilized weighted data provided by Ipsos to make the data nationally representative. Overall, 1189 individuals participated in the Wave 1 survey, and 843 participants completed the Wave 2 survey (71.6% retention rate). Those who intended to be vaccinated as soon as possible ("ASAP") were overwhelmingly vaccinated by Wave 2 (96%, 95% CI: 92% to 100%). Of those who initially wished to delay vaccination until there was more experience with it ("Wait and See"), 57% (95% CI: 47% to 67%) were vaccinated at Wave 2. Within the "Wait and See" cohort, those with income <$50,000 and those who had never received the influenza vaccine were significantly less likely to be vaccinated at Wave 2. Among those who initially indicated that they would not receive a COVID-19 vaccine ("Non-Acceptors"), 28% (95% CI: 21% to 36%) were vaccinated at Wave 2. Those who believed COVID-19 was not a major problem in their community were significantly less likely to be vaccinated, while those with more favorable attitudes toward vaccines in general and public health strategies to decrease the impact of COVID-19 were significantly more likely to be vaccinated. Overall, barriers to vaccine uptake for the "Wait and See" cohort appear to be more practical, whereas barriers for the "Non-Acceptor" cohort seem to be more ideological. These findings will help target interventions to improve uptake of COVID-19 boosters and future novel vaccines.

**Funding:** The study was supported in part by a research grant from Investigator-Initiated Studies Program of Merck Sharp & Dohme Corp grant awarded to Dr. Susan Rosenthal, Principal Investigator, and administered through Columbia University Irving Medical Center. The opinions expressed in this paper are those of the authors and do not necessarily represent those of Merck Sharp & Dohme Corp. The funders had no role in study design, data collection and analysis, decision to publish, or preparation of the manuscript.

**Competing interests:** Caroline Katzman, Julen Harris, Christine Mauro, Ariel de Roche, and Tucker Morgan have no conflicts of interest to declare. Gregory Zimet serves as an external advisory board member for Pfizer and Moderna, and as a consultant to Merck. He also has received investigator-initiated research funding from Merck administered through Indiana University and serves as an unpaid member of the Board of Directors for the Unity Consortium, a non-profit organization that promotes adolescent health through vaccination. Susan Rosenthal has received investigator-initiated research funding from Merck Investigator Studies Program administered through Columbia University Irving Medical Center. This does not alter our adherence to PLOS ONE policies on sharing data and materials.

## Introduction

Insufficient primary vaccine and booster uptake remains a major obstacle in curbing the severity and spread of SARS-CoV-2 infection [1, 2]. Historically, when a new vaccine is introduced, there is variability in uptake, with some individuals accepting the vaccine immediately and others being more cautious or refusing vaccination. The Diffusion of Innovation (DOI) model, often applied to public health promotion, has categorized this process by describing five adopter categories [3]. "Innovators," "Early Adopters," and the "Early Majority" tolerate more uncertainty and accept an intervention in the earlier stages of availability before the average person, whereas the "Late Majority" often require more education and time. In contrast, "Laggards" are the last to adopt an innovation and may reject it altogether. The DOI model can be applied to the rollout of COVID-19 vaccines. Many individuals sought vaccination as soon as possible despite logistical hurdles and may be categorized as "Innovators" and "Early Adopters". Meanwhile, the "Early Majority" waited for additional information, but chose to get vaccinated relatively soon after vaccination was approved for all adults. In contrast, the "Late Majority" waited for vaccination until there was more experience with the vaccines, and the "Laggards" declined vaccination despite a great deal more experience and knowledge regarding COVID-19 vaccination.

Several factors have been associated with delay or refusal of COVID-19 vaccination, including younger age, loss of income during the pandemic, a history of rejecting influenza vaccination, fear surrounding COVID-19 vaccine safety, negative attitudes toward the public health response to COVID-19, family and friends discouraging vaccination, concerns about shortcuts in vaccine development and profiteering, preference for "natural immunity," and political conservatism [4–8]. In addition, there seem to be different predictors of vaccination for those who are vaccinated earliest–the "Innovators," "Early Adopters," and "Early Majority." These individuals are more likely to have a household member over age 65, have received an influenza shot, have positive COVID-19 vaccine attitudes, and view COVID-19 vaccination as beneficial [9]. As suggested by the DOI model, COVID-19 vaccine hesitancy and refusal evolves over time, and many individuals who were initially hesitant to receive the vaccine prior to availability were vaccinated or willing to be vaccinated several months into vaccine rollout [10, 11]. Further, those who initially wished to "delay" vaccination–as opposed to those who "refused" vaccination–were significantly more likely to be vaccinated against COVID-19 by June 2021 [12].

It is unclear how attitudes early in vaccine availability impact vaccination longitudinally. This remains highly relevant given evolving recommendations about bivalent COVID-19 vaccine boosters and questions of annual or biannual COVID-19 vaccines in the future. Thus, there is a need to understand the factors that influence those who are initially vaccine hesitant or resistant, specifically the "Late Majority" and the "Laggards" to become vaccinated. Ultimately, identifying potential determinants of vaccine uptake over time is crucial to inform the development of public health strategies to increase acceptance of novel vaccines. Thus, we investigated demographics and attitudes toward COVID-19 and public health in the United States during the early stages of COVID-19 vaccine availability that were associated with vaccine uptake over the following 10 months, with a focus on those who were initially vaccine hesitant or resistant.

## Methods

### Study design and sampling strategies

Participants were members of Ipsos KnowledgePanel®, a probability-based web panel designed to be representative of the United States. KnowledgePanel has 60,000 adult

respondents, and members are sampled for each study separately. Panel members are recruited by Ipsos using an address-based sampling (ABS) methodology based on the latest Delivery Sequence File (DSF) of the United States Postal Service, with specific attention on hard-to-reach adults (i.e. those without internet) [13]. Ipsos uses a patented method to select active members based on benchmarks from the most recent U.S. Census to ensure the sample is weighted to be nationally representative. For this study, the target population consisted of non-institutionalized adults aged 18 and older residing in the U.S., and panel members were not excluded from a specific study based on other studies in which they are involved. Ipsos invited one adult representative sample of households to partake in the survey via email invitation. The surveys are completed online, and panelists who do not have internet access are provided with tablet devices and internet connection by Ipsos free of charge. The study authors developed the survey for the purposes of this study. Participants received a study information sheet stating that completion of the questionnaire indicated consent to participate. There were no potentially identifiable data gathered, and in fact, the authors only had access to de-identified data. The Institutional Review Board of Columbia University Irving Medical Center approved this study as an exempt protocol on December 22, 2020 (IRB-AAAT5154).

## Survey structure and relevant variables

The questionnaire has been described in detail in a previous publication on Wave 1 findings [9]. Questions asked in Wave 1 and Wave 2 surveys were highly similar, but the Wave 2 survey was updated to reflect the most up-to-date COVID guidelines and vaccine protocols. For each scale, items were administered randomly to minimize ordering effects.

**Sociodemographic variables.** In this study, sociodemographic variables included gender, age, race/ethnicity, income, education, political views, and U.S. region.

**General vaccine attitudes.** We assessed general vaccine attitudes through a scale that was created by calculating the mean across six questions [14, 15], including "I like the idea of vaccines," "vaccines are generally safe," "vaccines are a way to take good care of myself now and in the future," "vaccines are effective," "I get vaccinated because I can also protect people with a weaker immune system," and "'vaccination is a collective action to prevent the spread of diseases." Participants rated these questions on a 5-point Likert response scale from "strongly disagree" to "strongly agree". Additionally, participants were asked at Wave 1 if they had ever received an influenza vaccine.

**Attitudes and perceptions related to COVID-19 severity and prevention.** We used several categories of questions to assess attitudes and perceptions related to COVID-19 (S1 File). First, the survey included single items measuring if participants considered COVID-19 a major problem in their community, had a health condition making COVID-19 more severe, had a household member aged 65 years or older, and whether they had ever tested positive for COVID-19. Next, using a 5-point Likert response scale, participants rated seven items related to perceived severity of COVID-19 [16] (e.g., "I am scared about getting infected with COVID-19," rated from "strongly disagree" to "strongly agree"), eight items concerning the perceived effectiveness of behavioral strategies in protecting themselves and others from COVID-19 [17] (e.g.,"wearing a mask any time you leave the house to go out in public," rated from "not effective at all" to "extremely effective"), twelve items related to attitudes specifically toward the COVID-19 vaccine [18, 19] (e.g., "COVID-19 vaccines are important for my health", rated from "strongly disagree" to "strongly agree"), and eight reasons participants may decide to be vaccinated against COVID-19 (e.g., "getting a vaccine makes me personally less likely to get COVID-19", rated from "not at all important" to "very important"). The scale

items were evaluated in a prior study about attitudes toward COVID-19 vaccine uptake and were found to have good internal reliability [9].

## Data analysis

At the time of the Wave 1 survey (April 2021), COVID-19 vaccines were not yet available to all U.S. adults, thus, those who had received at least one dose of a vaccine or were planning to do so as soon as possible ("ASAP") were categorized as "Acceptors." The remaining participants were asked if they would like to receive the vaccine "after there is more experience with it". Those who responded "yes" were classified into the "Wait and See" category, while those who responded "no" were characterized as "Non-Acceptors." [9] Given that those in the "Wait and See" category and those in the "Non-Acceptor" category had differential rates of vaccination at Wave 2, the analyses were conducted separately for the two groups. At both waves, being vaccinated was defined as reporting receipt of at least one dose of a COVID-19 vaccine. Number of vaccines received by participants was not used in the analysis.

Participant demographics were summarized using weighted frequencies and percentages for each variable. For the purposes of this analysis, Wave 1 data (demographics, vaccination history, attitudes and perceptions related to COVID-19) were used to predict COVID-19 vaccination status at Wave 2. Each predictor variable was analyzed in a bivariate logistic regression model with vaccination status at Wave 2 as the outcome using SAS® Software v 9.4 (SAS Institute Inc., Cary, NC). Those who did not respond to the question of vaccination status at either wave were excluded from the analysis (Fig 1). Weighted values for Wave 2 data were calculated and provided by Ipsos for the findings to be nationally representative. The weights of Wave 1 respondents served as design weights for Wave 2 respondents, and these were adjusted to the geodemographic distributions from the 2019 American Community Survey.

There were three sources of missing data in this study. The first was missing data due to participants being loss-to-follow-up between Wave 1 and Wave 2. The Wave 2 weights provided by IPSOS and used in these analyses were adjusted to account for this. The second source of missing data were participants who did not complete the main item on vaccination in wave 2 (n = 12). Since this was such a small percentage of the overall sample, they were removed. The last source of missing data was on individual covariates of interest. These individuals were simply removed from the bivariate analyses as needed and never exceeded four observations lost. Because the number of participants in each of our subgroups were not predetermined, they are should be treated as random variables and the standard error estimates in the logistic regression models need to be adjusted to account for this additional source of variability. This was done by specifying the DOMAIN statement in proc surveylogistic in SAS [20].

## Results

### Demographics of the study sample

For "Wave 1" in April 2021, 1991 panelists were invited to participate in the study and 1189 participants completed the survey. For "Wave 2" in February 2022, 1062 panelists were invited to participate and 843 completed the survey (71.6% retention rate). Overall, of the 843 participants retained at Wave 2, over half were male and Non-Hispanic white. The sample was older with over 70% of participants over age 45. The population was also more educated, with over half having attended at least some college. For a full listing of non-weighted and weighted descriptive variables of the sample, see Table 1.

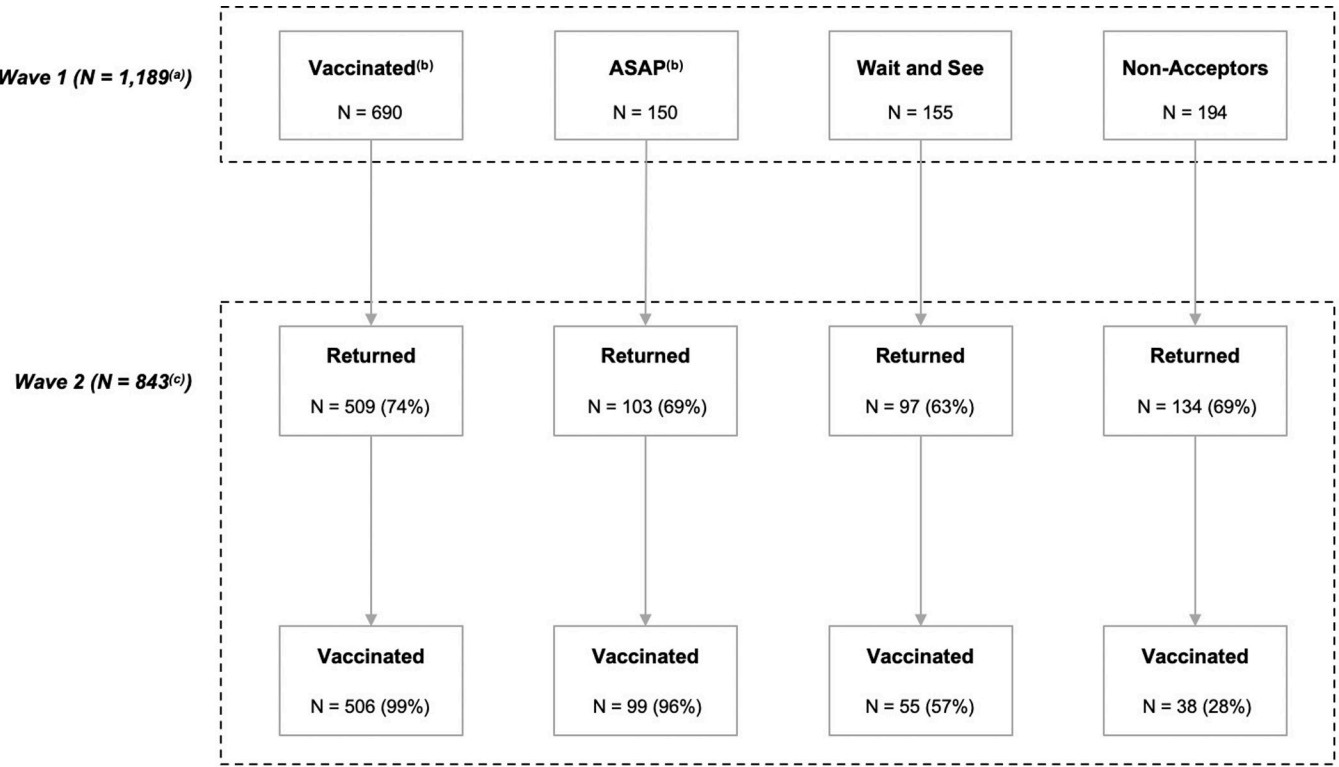

**Fig 1. COVID-19 vaccine status of participants at Wave 1 & Wave 2.** (a) Excludes 19 participants who did not answer the question of vaccination status and were not included in the analysis (b) "Vaccinated" and "ASAP" groups were combined for the analysis (c) Excludes 12 participants who returned but did not answer the question of vaccination status.

### Vaccine uptake at Wave 1 vs. Wave 2

In Fig 1, we present vaccination status from Wave 1 to Wave 2. Briefly, of the 509 participants who were vaccinated at Wave 1 and returned for follow-up, all but 3 reported being vaccinated by Wave 2 (99%, 95% CI: 99% to 100%), this was likely due to participant error and is not significant. Of those 103 participants who stated they would like to be vaccinated "ASAP" and returned, all but 4 were vaccinated at Wave 2 (96%, 95% CI: 92% to 100%),). As virtually all participants who wished to be vaccinated "ASAP" at Wave 1 had received the COVID-19 vaccine by Wave 2, this validated classification of the "ASAP" participants as "Acceptors" at Wave 1. Given the very high acceptance rates, factors impacting vaccine uptake for acceptors and ASAP participants could not be analyzed.

Of the 155 participants who stated at Wave 1 that they would make decisions about vaccination when they had more information ("Wait and See"), 97 participants returned and 55 of these individuals had received a vaccine by Wave 2 (57%, 95% CI: 47% to 67%),). Of the 134 participants who were classified as "Non-Acceptors" and returned, 38 participants had been vaccinated by Wave 2 (28%, 95% CI: 21% to 36%)).

### "Wait and See"

In the bivariate analysis (Table 2), those classified as "Wait and See" who were in the <$50,000 income category had significantly lower odds of receiving the COVID-19 vaccine at Wave 2 compared to respondents in the >$200,000 income category (OR = 0.21, 95%CI = 0.05–0.88). Additionally, respondents who had never received the influenza vaccine were significantly less

**Table 1. Demographics of participants in total sample (N = 843).**

| Demographics | Raw *n* | Weighted *n* (%) |
|---|---|---|
| **Gender** | | |
| Male | 452 | (48.6%) |
| Female | 391 | *432.5 (51%)* |
| **Age (years)** | | |
| 60+ | 359 | (29.9%) |
| 45–59 | 239 | (24.8%) |
| 30–44 | 165 | (25.3%) |
| 18–29 | 80 | 168.6 (20.0%) |
| **Race/Ethnicity** | | |
| Non-Hispanic White | 609 | (63.3%) |
| Non-Hispanic Black | 83 | (11.7%) |
| Hispanic | 86 | (16.2%) |
| 2+ races or other, non-Hispanic | 72 | 73.6 (8.7%) |
| **Annual Income** | | |
| More than $150,000 | 210 | (20.0%) |
| $100,000–149,999 | 164 | (19.1%) |
| $75,000–99,999 | 122 | (14.2%) |
| $50,000–74,999 | 149 | (17.1%) |
| Less than $50,000 | 198 | 249.0 (29.6%) |
| **Education** | | |
| Bachelor's degree or higher | 308 | (31.6%) |
| Some college | 255 | (30.0%) |
| High school degree or less | 280 | 323.5 (38.4%) |

likely to receive the COVID-19 vaccine at Wave 2 than those who had received the influenza vaccine in the past (OR = 0.29, 95%CI = 0.11–0.77). Other socio-demographic variables and items measuring attitudes toward vaccination and COVID-19 were not predictive of COVID-19 vaccination at Wave 2 for the "Wait and See" category.

## "Non-Acceptors"

The bivariate analysis for those who were classified as "Non-Acceptors" (Table 3) demonstrated that those who had not viewed COVID-19 as a major problem in their community had significantly lower odds of vaccination by Wave 2 (OR = 0.30, 95%CI = 0.12–0.76). Those COVID-19 vaccine "Non-Acceptors" who, at Wave 1, considered behavioral health strategies to be more effective in protecting oneself or others (OR = 2.43, 95%CI = 1.51–3.93), had more favorable general vaccine attitudes (OR = 2.45, 95%CI = 1.17–5.14), had more favorable COVID-19 vaccine attitudes (OR = 2.76, 95%CI = 1.45–5.25) and had greater perceived benefits of COVID-19 vaccines (OR = 2.26, 95%CI = 1.43–3.56) were significantly more likely to be vaccinated at Wave 2.

## Discussion

This longitudinal study investigated whether COVID-19 vaccine intentions, attitudes, and perceptions at Wave 1, in April 2021, predicted vaccine uptake ten months later at Wave 2, in February 2022. Overall, findings demonstrate that plans for COVID-19 vaccination in the early stages of vaccine availability were predictive of vaccination status one year later. In early 2021, there were numerous barriers to COVID-19 vaccination in the U.S., including eligibility,

**Table 2. Demographic and health-related characteristics of survey respondents (Wait and See), bivariate logistic regression models for vaccine uptake (N = 97)[1].**

| Categorical Variables | Raw n | Weighted n (%) | Bivariate Unadjusted OR [95% CI] Received Vaccine vs. Did Not Receive |
|---|---|---|---|
| **Wave 2 Vaccination Status** | | | |
| Received Vaccine | 55 | 62.55 (58.2%) | -- |
| Did Not Receive Vaccine | 42 | 44.66 (41.7%) | -- |
| **Gender** | | | |
| *Male* | 55 | 51.8 (48.3%) | Ref |
| Female | 42 | 55.4 (51.7%) | 2.26 [0.90, 5.67] |
| **Age (years)** | | | |
| *60+* | 31 | 23.8 (22.2%) | Ref |
| 45–59 | 26 | 23.5 (21.9%) | 0.90 [0.30, 2.71] |
| 30–44 | 30 | 40.1 (37.4%) | 1.38 [0.47, 4.12] |
| 18–29 | 10 | 19.9 (18.5%) | 2.04 [0.41, 10.12] |
| **Race/Ethnicity** | | | |
| *Non-Hispanic White* | 65 | 67.1 (62.6%) | Ref |
| Non-Hispanic Black | 13 | 16.4 (15.3%) | 0.73 [0.20, 2.66] |
| Hispanic | 10 | 16.0 (14.9%) | 0.81 [0.19, 3.42] |
| 2+ races or other, non-Hispanic | 9 | 7.8 (7.3%) | 0.68 [0.15, 3.09] |
| **Annual Income** | | | |
| *More than $150,000* | 16 | 13.9 (13.0%) | Ref |
| $100,000–149,999 | 19 | 19.6 (18.3%) | 0.22 [0.05, 1.08] |
| $75,000–99,999 | 12 | 13.7 (12.7%) | 0.94 [0.16, 5.43] |
| $50,000–74,999 | 22 | 22.5 (21.0%) | 0.41 [0.09, 1.92] |
| Less than $50,000 | 28 | 37.5 (35.0%) | **0.21 [0.05, 0.88]*** |
| **Education** | | | |
| *Bachelor's degree or higher* | 21 | 16.6 (15.4%) | Ref |
| Some college | 41 | 45.8 (42.7%) | 0.93 [0.29, 2.97] |
| High school degree or less | 35 | 44.8 (41.8%) | 0.50 [0.16, 1.60] |
| **Political views** | | | |
| *Very Liberal/Liberal* | 12 | 16.9 (16.0%) | Ref |
| Moderate/Middle of the Road | 35 | 46.2 (43.7%) | 1.56 [0.38, 6.41] |
| Very Conservative/ Conservative | 43 | 36.2 (34.3%) | 1.05 [0.27, 4.07] |
| Prefer not to answer | 6 | 6.3 (6.0%) | 3.34 [0.41, 27.10] |
| **Region of Country** | | | |
| *Northeast* | 18 | 21.7 (20.3%) | Ref |
| Midwest | 25 | 27.5 (25.6%) | 0.91 [0.22, 3.84] |
| South | 35 | 38.2 (35.7%) | 0.71 [0.19, 2.66] |
| West | 19 | 19.8 (18.4%) | 0.55 [0.12, 2.44] |
| **Household member ≥ age 65** | | | |
| *Yes* | 28 | 23.7 (22.4%) | Ref |
| No | 68 | 82.3 (77.6%) | 1.40 [0.54, 3.63] |
| **Ever had a flu vaccine** | | | |
| *Yes* | 59 | 64.0 (59.7%) | Ref |
| No | 38 | 43.2 (40.3%) | **0.29 [0.11, 0.77]*** |
| **Health condition making COVID-19 more severe** | | | |
| *Yes* | 30 | 33.6 (31.3%) | Ref |
| No & Not Sure | 67 | 73.6 (68.6%) | 0.89 [0.34, 2.33] |
| **Tested positive for COVID-19** | | | |

*(Continued)*

**Table 2.** (Continued)

| Categorical Variables | Raw *n* | Weighted *n* (%) | Bivariate Unadjusted OR [95% CI] Received Vaccine vs. Did Not Receive |
|---|---|---|---|
| *Yes* | 22 | 25.6 (23.9%) | Ref |
| No | 51 | 55.9 (52.2%) | 0.57 [0.18, 1.79] |
| Not sure | 24 | 25.7 (24.0%) | 0.60 [0.16, 2.24] |
| **View COVID-19 as a major problem in community** | | | |
| *Yes* | 37 | 44.3 (41.3%) | Ref |
| No | 60 | 62.9 (58.7%) | 0.97 [0.38, 2.43] |
| Scale Variables | *n* obs | Mean Item Score (SE) | Bivariate Unadjusted OR [95% CI] Received Vaccine vs. Did Not Receive |
| **Perceived COVID severity** | 97 | 2.74 (0.11) | 1.01 [0.60, 1.71] |
| **Effectiveness of behavioral strategies to protect self/others** | 97 | 3.60 (0.10) | 1.25 [0.74, 2.10] |
| **General vaccine attitudes** | 95 | 3.53 (0.08) | 1.18 [0.52, 2.67] |
| **COVID vaccine attitudes** | 95 | 2.83 (0.06) | 1.60 [0.57, 4.49] |
| **COVID vaccine reasons** | 96 | 3.01 (0.10) | 1.13 [0.66, 1.92] |

[1]Bolded values are significant at

*p<0.05

availability, and scheduling. Despite these challenges, those who stated at Wave 1 that they would like to be vaccinated as soon as possible did overwhelmingly receive a vaccine by Wave 2. Thus, for this cohort, intention to vaccinate was a valid predictor of future behavior.

The DOI model suggests that more individuals will accept a public health intervention over time. Our data support the application of this model to COVID-19 vaccines, and add to evidence to the conclusion that both vaccine hesitancy and refusal are not fixed, but rather, often change over time [5, 6, 11]. In this longitudinal study, we found that individuals who wished to wait until there was more experience with the COVID-19 vaccine at Wave 1 –early in vaccine availability–were significantly more likely to be vaccinated at Wave 2 (57%) than those who did not want to be vaccinated even with more experience with the vaccine (28%). This finding aligns with prior studies demonstrating that individuals who preferred to delay COVID-19 vaccination are significantly more likely to be vaccinated than those who initially refused it [12]. Additionally, although a majority of "Non-Acceptors" were not vaccinated at Wave 2, a surprising and substantial minority did receive the vaccine.

Undoubtedly, there were myriad factors that may have led individuals to get vaccinated between Wave 1 and Wave 2 –including vaccine mandates, personal experiences, employment pressures, and SARS-CoV-2 infections. While we cannot say with certainty which specific factors led participants in the "Wait and See" and "Non-Acceptor" cohorts to receive the COVID-19 vaccine, this study provides crucial insight into several measures that were significantly predictive of vaccination over a longitudinal period. These findings remain important and timely with the addition of the bivalent COVID-19 vaccines and boosters, as well as the possibility of recommended annual COVID-19 vaccination. As the landscape of SARS-CoV-2 immunity evolves, understanding factors that predict vaccine uptake should guide the creation of public health campaigns and support effective patient vaccine counseling during individual clinical encounters.

For participants in the "Wait and See" cohort at Wave 1, those with lower incomes were significantly less likely to be vaccinated at Wave 2 than those with the highest incomes. There were no significant differences in vaccine uptake at Wave 2 for the other income brackets. This

**Table 3. Demographic and health-related characteristics of survey respondents (Non-Acceptors), bivariate logistic regression models for vaccine uptake (N = 134)[1].**

| Categorical Variables | Raw n | Weighted n (%) | Bivariate Unadjusted OR [CI 95%] Received Vaccine vs. Did Not Receive |
|---|---|---|---|
| **Wave 2 Vaccination Status** | | | |
| Received Vaccine | 38 | 48.0 (30.5%) | -- |
| Did Not Receive Vaccine | 96 | 109.4 (69.5%) | -- |
| **Gender** | | | |
| *Male* | 73 | 79.0 (50.2%) | Ref |
| Female | 61 | 78.4 (49.8%) | 0.63 [0.25, 1.56] |
| **Age (years)** | | | |
| *60+* | 34 | 23.8 (15.5%) | Ref |
| 45–59 | 38 | 32.4 (20.6%) | 1.77 [0.58, 5.38] |
| 30–44 | 44 | 58.3 (37.0%) | 1.26 [0.40, 3.92] |
| 18–29 | 18 | 42.2 (26.8%) | 2.53 [0.66, 9.67] |
| **Race/Ethnicity** | | | |
| *Non-Hispanic White* | 103 | 104.4 (66.3%) | Ref |
| Non-Hispanic Black | 14 | 26.7 (16.9%) | 3.45 [0.94, 12.63] |
| Hispanic | 13 | 22.8 (14.5%) | 1.48 [0.37, 5.97] |
| 2+ races or other, non-Hispanic | 4 | 3.5 (2.2%) | 5.48 [0.56, 53.66] |
| **Annual Income** | | | |
| *More than $150,000* | 23 | 17.3 (11.0%) | Ref |
| $100,000–149,999 | 25 | 30.4 (19.3%) | 0.79 [0.22, 2.81] |
| $75,000–99,999 | 16 | 14.9 (9.5%) | 0.93 [0.21, 4.18] |
| $50,000–74,999 | 22 | 24.9 (15.8%) | 0.43 [0.10, 1.92] |
| Less than $50,000 | 48 | 69.9 (44.4%) | 0.48 [0.15, 1.54] |
| **Education** | | | |
| *Bachelor's degree or higher* | 22 | 20.4 (13.0%) | Ref |
| Some college | 47 | 53.0 (33.6%) | 0.41 [0.12, 1.40] |
| High school degree or less | 65 | 84.0 (53.4%) | 0.49 [0.15, 0.16] |
| **Political views** | | | |
| *Very Liberal/Liberal* | 11 | 16.3 (10.5%) | Ref |
| Moderate/Middle of the Road | 35 | 43.8 (28.3%) | 0.26 [0.06, 1.17] |
| Very Conservative/ Conservative | 67 | 65.3 (42.2%) | 0.28 [0.07, 1.10] |
| Prefer not to answer | 19 | 29.4 (19.0%) | 0.29 [0.05, 1.80] |
| **Region of Country** | | | |
| *Northeast* | 19 | 18.7 (11.9%) | Ref |
| Midwest | 30 | 39.4 (25.0%) | 0.84 [0.19, 3.65] |
| South | 60 | 68.2 (43.3%) | 0.70 [0.180, 2.76] |
| West | 25 | 31.1 (19.8%) | 1.05 [0.24, 4.64] |
| **Household member ≥ age 65** | | | |
| *Yes* | 40 | 39.3 (24.9%) | Ref |
| No | 94 | 118.1 (75.1%) | 2.02 [0.75, 5.41] |
| **Ever had a flu vaccine** | | | |
| *Yes* | 58 | 63.6 (40.4%) | Ref |
| No | 76 | 93.8 (59.6%) | 0.43 [0.18, 1.03] |
| **Health condition making COVID-19 more severe** | | | |
| *Yes* | 25 | 26.1 (16.6%) | Ref |
| No & Not Sure | 109 | 131.3 (83.4%) | 0.60 [0.21, 1.71] |
| **Tested positive for COVID-19** | | | |

*(Continued)*

**Table 3.** (Continued)

| Categorical Variables | Raw *n* | Weighted *n* (%) | Bivariate Unadjusted OR [CI 95%] Received Vaccine vs. Did Not Receive |
|---|---|---|---|
| *Yes* | 38 | 47.0 (30.1%) | Ref |
| No | 78 | 91.2 (58.4%) | 0.40 [0.15, 1.06] |
| Not sure | 17 | 18.0 (11.5%) | 0.31 [0.06, 1.56] |
| **View COVID-19 as a major problem in community** | | | |
| *Yes* | 36 | 49.2 (31.4%) | Ref |
| No | 97 | 107.6 (68.6%) | **0.30 [0.12, 0.76]** |
| **Scale Variables (MEANS)** | **n obs** | **Mean Item Score (SE)** | **Bivariate Unadjusted OR [CI 95%], Received Vaccine vs. Did Not Receive** |
| **Perceived COVID severity** | 134 | 2.20 (0.10) | 1.43 [0.94, 2.18] |
| **Effectiveness of behavioral strategies to protect self/others** | 134 | 3.06 (0.11) | **2.43 [1.51, 3.93]**\*\* |
| **General vaccine attitudes** | 133 | 2.92 (0.08) | **2.45 [1.17, 5.14]**\* |
| **COVID vaccine attitudes** | 130 | 2.28 (0.08) | **2.76 [1.45, 5.25]**\*\* |
| **COVID vaccine reasons** | 130 | 2.06 (0.11) | **2.26 [1.43, 3.56]**\*\* |

[1]Bolded values are significant at

\*$p < 0.05$

\*\*$p < 0.01$

fits with existing literature that those in lower income brackets are less likely to receive vaccines and general preventative care than those in higher income groups [21]. This finding has also been shown specifically for the COVID-19 vaccine [4, 22]. Additionally, structural and logistical barriers–such as travel time to clinic, childcare needs, limited access to preventative care, and lost wages due to time off for vaccination–have previously been identified as factors contributing to completion of the hepatitis B vaccine series [23]. For both hepatitis B and COVID-19 vaccination, income is likely to be an indicator of difficulty overcoming structural and logistical barriers.

Further, those in the "Wait and See" category who had received an influenza vaccine in the past were significantly more likely to be vaccinated at Wave 2 than those who had never been vaccinated for influenza. While it can be assumed that this cohort was less resistant to vaccination in general, we found that attitudes toward vaccination were not actually predictive of vaccination status in this group. Thus, it is more likely that these participants were more easily able to overcome the practical barriers to vaccination; perhaps their past influenza vaccination indicates that they were better able to navigate health systems, or that they were able to access preventative care measures more easily. Previous studies have shown that past influenza vaccination is a strong predictor of future influenza vaccination–which in part is due to ability to access care [24]. Again, this demonstrates that pragmatic and logistical barriers may be an important factor impacting vaccination status for the "Wait and See" cohort. Further, while this study focuses on predictors of novel vaccine uptake, utilizing strategies from successful influenza vaccination campaigns may be increasingly useful given ongoing discussions of annual COVID-19 booster vaccines, especially with the possibility of combined influenza/COVID-19 vaccines in the future.

It was initially surprising that attitudes toward vaccines, COVID-19, and public health measures were not predictive of COVID-19 vaccine status in the "Wait and See" cohort, as we anticipated this would be a driver for these participants given prior literature on vaccine attitudes and hesitancy [7]. However, it is worth noting that this group was not "vaccine negative"

overall. Instead, they were individuals who wanted more information before receiving the vaccine. Overall, this cohort was presumably more open-minded toward vaccination, and negative attitudes were likely less relevant to their decision-making process, corroborating our hypothesis that barriers to vaccination were more likely practical–as opposed to ideological–for this group. This is a novel finding that has widespread implications for vaccine counseling and outreach.

In contrast, for COVID-19 vaccine "Non-Acceptors," Wave 1 attitudes were predictive of vaccination status at Wave 2 –those with less favorable attitudes toward public health measures, vaccination in general, and less concern about COVID-19 were significantly less likely to be vaccinated at Wave 2. This set of findings supports prior data which indicates that worry about vaccination is associated with lower vaccine uptake [4]. It is likely that negative attitudes at Wave 1 were a crucial factor preventing vaccination for this group. Additionally, those "Non-Acceptors" with more favorable public health attitudes to begin with may have been more accepting of influences such as vaccine mandates and social pressures. While vaccine "Non-Acceptors" at Wave 1 were significantly less likely than the "Wait and See" cohort to be vaccinated at Wave 2, almost 30% of the participants who returned did end up receiving the vaccine. We hypothesize that this percentage could be even higher with targeted interventions, which is an important area for future research.

We have demonstrated that those who want to be vaccinated "ASAP" will follow through on this intention. Practical and structural barriers can play a critical role in access to vaccines, particularly for those who are initially vaccine hesitant and are waiting for more experience with the vaccine, and for those with less economic resources. Overall, investing time and resources into dismantling logistical barriers for these individuals will be a valuable investment of public health resources. Additionally, it may be useful to develop targeted counseling messages specifically for those who have received the influenza vaccine, as these individuals may be particularly cautious about novel vaccines, but not resistant to all vaccines. Individuals who received influenza vaccines in the past likely have more comfort with navigating the healthcare system, and different counseling strategies may be helpful in improving comfort with new medical advancements.

For vaccine "Non-Acceptors," barriers to vaccination seem to be more ideological than practical. For these individuals, understanding which educational strategies will address their vaccine concerns (both in general and regarding novel vaccines) will be useful in deciding how best to spend public health resources and provider time. In addition, the results suggest that many "Non-Acceptors" may change their minds over time and should not be ignored when implementing strategies to improve vaccination rates.

## Limitations

There were several limitations to this study. First, our sample size for this analysis was relatively small given the number of participants who had been previously vaccinated at Wave 1. However, this was partially overcome by weighting our data to be nationally representative. Additionally, given the unique social environment, political context, and resource availability in the U.S., these findings are likely not applicable to other countries. Moreover, our study was designed to be nationally representative, though not regionally representative, so it is unknown how findings may have varied within different U.S. regions based on diverse social and political climates. Further, it is impossible to know the role that mandates played in vaccination of the participants.

In addition to mandates, there are many other factors that may have contributed to vaccine uptake that were not investigated–including new infections, job requirements, and family

illnesses and deaths. Additionally, vaccination status was self-reported. Lastly, we defined vaccination as having received at least one dose of a vaccine, and it is possible that those who initially accepted the vaccine did not complete the series, despite current understanding that receiving two doses of a primary mRNA series and recommended boosters is important in preventing serious outcome or death [25].

## Conclusion

The COVID-19 pandemic has been a foundational moment in the crafting of future emergency public health responses. In a resource-scarce and time-limited environment, it is crucial to consider how best to target public health campaigns for the widest impact. Recognizing that people will fall on a continuum of vaccine acceptance could help accelerate uptake by targeting interventions to the different groups as quickly and precisely as possible [3]. We identified a number of factors that are correlated with increased COVID-19 vaccine uptake over a longitudinal period. Overall, both structural and ideological barriers likely play a role in vaccine uptake and understanding where individuals fall on the continuum can help to target interventions and increase vaccination.

## Supporting information

**S1 File. Selected scale items from Wave 1 questionnaire.**
(DOCX)

## Acknowledgments

**Consent to participate**

All participants in this study are members of the IPSOS KnowledgePanel®. Participants received an information sheet for this study which stated that completion of the questionnaire indicated consent to participate in this study.

**Consent for publication**

There is no potentially identifiable data presented, and in fact, the authors only had access to de-identified data.

## Author Contributions

**Conceptualization:** Julen Harris, Gregory Zimet, Susan Rosenthal.

**Data curation:** Tucker Morgan, Ariel de Roche, Christine Mauro.

**Formal analysis:** Tucker Morgan, Ariel de Roche, Christine Mauro.

**Methodology:** Susan Rosenthal.

**Supervision:** Gregory Zimet, Susan Rosenthal.

**Visualization:** Gregory Zimet.

**Writing – original draft:** Caroline Katzman, Tucker Morgan.

**Writing – review & editing:** Caroline Katzman, Julen Harris.

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
