## [Decision Letter · Decision Letter 0]

22 May 2023

PONE-D-23-06419Longitudinal Assessment of COVID-19 Vaccine Uptake: A Two-Wave Survey of a Nationally Representative U.S. SamplePLOS ONE

Dear Dr. Katzman,

Thank you for submitting your manuscript to PLOS ONE. After careful consideration, we feel that it has merit but does not fully meet PLOS ONE’s publication criteria as it currently stands. Therefore, we invite you to submit a revised version of the manuscript that addresses the points raised during the review process.

We look forward to receiving your revised manuscript.

Kind regards,

Jerome Nyhalah Dinga, PhD

Academic Editor

PLOS ONE

“The study was supported in part by a research grant from Investigator-Initiated Studies Program of Merck Sharp & Dohme Corp grant awarded to Dr. Susan Rosenthal, Principal Investigator, and administered through Columbia University Medical Center. The opinions expressed in this paper are those of the authors and do not necessarily represent those of Merck Sharp & Dohme Corp.” 

“Funding

The study was supported in part by a research grant from Investigator-Initiated Studies Program of Merck Sharp & Dohme Corp grant awarded to Dr. Susan Rosenthal, Principal Investigator, and administered through Columbia University Medical Center. The opinions expressed in this paper are those of the authors and do not necessarily represent those of Merck Sharp & Dohme Corp.”

“The study was supported in part by a research grant from Investigator-Initiated Studies Program of Merck Sharp & Dohme Corp grant awarded to Dr. Susan Rosenthal, Principal Investigator, and administered through Columbia University Medical Center. The opinions expressed in this paper are those of the authors and do not necessarily represent those of Merck Sharp & Dohme Corp.” 

“Caroline Katzman, Julen Harris, Christine Mauro, Ariel de Roche, and Tucker Morgan have no conflicts of interest to declare. Gregory Zimet serves as an external advisory board member for Pfizer and Moderna, and as a consultant to Merck. He also has received investigator-initiated research funding from Merck administered through Indiana University and serves as an unpaid member of the Board of Directors for the Unity Consortium, a non-profit organization that promotes adolescent health through vaccination. Susan Rosenthal has received investigator-initiated research funding from Merck Investigator Studies Program administered through Columbia University Irving Medical Center.”

7. Your ethics statement should only appear in the Methods section of your manuscript. If your ethics statement is written in any section besides the Methods, please move it to the Methods section and delete it from any other section. Please ensure that your ethics statement is included in your manuscript, as the ethics statement entered into the online submission form will not be published alongside your manuscript.

8. Please include a caption for figure 1.

9. Please include your tables as part of your main manuscript and remove the individual files. Please note that supplementary tables (should remain/ be uploaded) as separate "supporting information" files

Reviewers' comments:

Reviewer's Responses to Questions

**Comments to the Author**

1. Is the manuscript technically sound, and do the data support the conclusions?

Reviewer #1: No

Reviewer #2: Yes

2. Has the statistical analysis been performed appropriately and rigorously? 

Reviewer #1: No

Reviewer #2: Yes

3. Have the authors made all data underlying the findings in their manuscript fully available?

Reviewer #1: No

Reviewer #2: Yes

4. Is the manuscript presented in an intelligible fashion and written in standard English?

Reviewer #1: Yes

Reviewer #2: Yes

5. Review Comments to the Author

Reviewer #1: This survey conducted in the United States adds to the evidence base concerning factors that drive one of the biggest global public health threats, i.e. vaccine hesitancy. Unfortunately, the study population was very small, resulting in highly imprecise estimates. Nonetheless, the study is interesting, and I especially enjoyed how the authors applied the “Diffusion of Innovation” model to how people behave with respect to COVID-19 vaccines.

That said, I believe the manuscript would benefit from revisions in order to more clearly and transparently describe various methodological aspects as well as the study findings. I hope my comments and suggestions below will help improve the paper.

ABSTRACT:

1. I strongly suggest briefly describing the study methods instead of jumping directly from rationale to results. In doing so, please make sure to clearly define the population of interest (adults? children? older adults? others?), indicate where the study was conducted, and describe participants’ selection methods (are these individuals meants to be representative of the general population?).

2. Please indicate the size the survey population and whether/how this changed from Wave 1 to Wave 2. It’s hard to interpret the study findings without any idea of how many people participated. Relatedly, I would suggest including 95% confidence intervals of prevalence proportions reported here.

INTRODUCTION:

1. As I am sure you know, COVID-19 is the disease caused by SARS-CoV-2, so the expression “COVID-19 infection” is not entirely correct and should be replaced with “SARS-CoV-2 infection”. Please revise the terminology throughout the manuscript to refer to COVID-19 or SARS-CoV-2 as appropriate.

2. Lines 64-66: I understand this may be intuitive, but I would suggest citing one or more references to support this statement.

3. Line 83: Consider adding the following citation along with references 2-5: Basta NE et al. Factors Associated With Willingness to Receive a COVID-19 Vaccine Among 23,819 Adults Aged 50 Years or Older: An Analysis of the Canadian Longitudinal Study on Aging. Am J Epidemiol. 2022 May 20;191(6):987-998. doi: 10.1093/aje/kwac029.

4. Lines 92-94: I would suggest rephrasing this sentence for clarity.

5. Lines 96-99: As also mentioned with respect to the abstract, I do think that the population and country of study should be clearly indicated. I encourage you to revise this last section of the introduction to incorporate this information. While some factors certainly play a role in favoring or hindering vaccine uptake across countries, other are more context specific. Given the peculiar social and political climate of the United States, I believe it is important to highlight the study context early on, as this also affects methodological choices as well as the interpretation of findings.

METHODS:

1. Consider slightly reorganizing the methods section into sub-sections with their own sub-headings to more systematically describe: the study design, cohort characteristics/sampling strategies, data collection methods, survey structure and relevant variables, and data analysis.

2. I am not familiar with this Ipsos panel, and it is unclear to me whether the survey was paper-based, by phone or online. Or perhaps multiple options were offered to participants to accommodate their needs and preferences?

3. What is the size of the Ipsos survey panel?

4. Considering that you leveraged on an existing survey panel, can you please clarify if the survey described in this article was developed ad hoc by the study authors or was part of a larger survey covering other health or social aspects?

5. If I understand correctly, Ipsos participants were invited to participate to your Waves 1 and 2 in addition to other surveys they were already involved in. Did you sample a subgroup of Ipsos participants or invited all of them to complete the questionnaire?

6. For clarity and transparency, it would be helpful to see the list of questions, response options along with the resulting variables and their categorization. For example, you mention gender, race/ethnicity, political views, and other variables, but we don’t know how this was categorized. I would suggest including this information in a supplementary file. However, I do think that some key details should be explained in-text. For example, the definition of “general vaccine attitudes” should be more clearly described considering that the study is focused around this concept. So, knowing which six questions were used to calculate this score is extremely important to the reader. Consider summarizing this information in a table.

7. Please clarify whether questionnaires administered at Wave 1 and Wave 2 include the exact same questions.

8. Please explain how missing data were handled.

9. How did you calculate weights?

10. How did you account for “variability introduced by subsampling”? Please expand the statistical analysis sub-section to provide some more details.

RESULTS:

1. As noted above, I think there is not enough clarity around the size of the study population. From Figure 1, we know that 1,189 people completed the Wave 1 survey and 843 completed the Wave 2 survey. It is still unclear what proportion of Ipsos participants were initially contacted and invited. Also, even if the numbers above are available in the flow diagram, I think it is important to include this information in-text as well, ideally in the Results’ opening sentence.

2. I am quite surprised to see that only regression analysis results are presented in the article. Please include descriptive statistics (table and text) before heading to the results to the logistic regression model.

3. Lines 162-164: I find this sentence quite confusing: “Briefly, of the 509 participants who were vaccinated at Wave 1 and returned for follow-up, all but 3 reported being vaccinated by Wave 2 (99%).” Is these people were vaccinated at Wave 1, what were they supposed to report at Wave 2? Did you ask them if they had received a second dose? Or just to confirm their vaccination status as reported at Wave 1? Please rephrase and clarify.

4. From the description of the results, I have trouble understanding whether there was any attempt to differentiate between first, second and booster doses. I believe this is an important point to discuss in the manuscript.

DISCUSSION:

1. I appreciate the discussion around the meaning and implications of the study findings. However, I strongly encourage you to revise it in order to incorporate considerations on whether and how your findings relate to other studies. What’s new in this study that was not captured in other studies? Do you think that your findings apply to other countries and why or why not? Also, what do your results tell us about attitudes towards vaccination in different parts of the US? I know the size of the study population was quite small and you certainly lacked power to conduct any subgroup analyses, but – in my opinion – this is an important point to discuss.

Reviewer #2: GENERAL COMMENTS TO AUTHORS

Overall this is a relatively well-designed and implemented study involving two KnowledgePanel surveys regarding COVID-19 vaccination intent and receipt- one fielded in April 2021 (when only some Americans were recommended to be vaccinated n the staged-in process due to vaccine supply) and the follow-up survey in February 2022 (when all Americans were vaccine-eligible). The study goal was to identify predictors of ultimate vaccination. The authors used a reasonable Diffusion of Innovation (DOI) conceptual model to ground their study and analysis. They found, unsurprisingly, that those who wanted the vaccine “ASAP” nearly all received it later, those who were cleary non-acceptors initially remained so afterwards, and those in between tended to be vaccinated with higher-income and barriers being moderate predictors in the wait-and-see group while belief structures were paramount in the non-acceptor group.

The study was done well, and the findings are believable. Key limitations are that Wave 1 occurred when some could get vaccinated so even the initial group was unusual (for example leaving out healthcare, essential workers, and older adults) from eligibility. Thus even prediction suffers from an unusual initial group. Second, sample sizes per DOI group were small (between 100-200 mostly), limiting confidence in prediction. Third, with the rapidly shifting background and concerns having to do with receipt of bivalent vaccine in a population which has largely already had COVID and largely already had at least 1 vaccination, the study implications at this point are limited.

SPECIFIC COMMENTS

Introduction

While well written, this section would be enhanced by relating the DOI and concepts to our current situation regarding COVID-19 vaccination and the likely future situation of having annual or biannual vaccine recommendations.

Methods

In general the methods were described well.

The authors could include a bit more on the underlying KnowledgePanel methods (recognizing that they published results of their original wave 1 study.

Results

The findings are presented relatively well and clearly

Discussion

This section is well thought out. It could be improved by increasing the discussion of study findings as it relates to the current situation of the US recommending bivalent vaccination for a population that has largely had COVID infection and has had at least one prior vaccination.

The discussion about relationship to influenza vaccination is not new and does not currently lend itself to actions. The authors should consider discussing how knowledge of influenza vaccination might practically help providers or health systems in their attempt to promote COVID-19 vaccination (beyond just showing the association). In addition, future COVID-19 vaccines might be combined with influenza vaccines so a discussion could cover that topic.

---

## [Author Response · Author response to Decision Letter 0]

6 Jul 2023

We have edited this paper to fully reflect PLOS ONE’s style requirements including those for file naming. 

All participants in this study are members of the IPSOS KnowledgePanel®. Participants received an information sheet for this study which stated that completion of the questionnaire indicated consent to participate in this study. This has been added to the Methods and the Ethics Statement.

“The study was supported in part by a research grant from Investigator-Initiated Studies Program of Merck Sharp & Dohme Corp grant awarded to Dr. Susan Rosenthal, Principal Investigator, and administered through Columbia University Medical Center. The opinions expressed in this paper are those of the authors and do not necessarily represent those of Merck Sharp & Dohme Corp.” 

We have added this statement. 

“Funding

The study was supported in part by a research grant from Investigator-Initiated Studies Program of Merck Sharp & Dohme Corp grant awarded to Dr. Susan Rosenthal, Principal Investigator, and administered through Columbia University Medical Center. The opinions expressed in this paper are those of the authors and do not necessarily represent those of Merck Sharp & Dohme Corp.”

“The study was supported in part by a research grant from Investigator-Initiated Studies Program of Merck Sharp & Dohme Corp grant awarded to Dr. Susan Rosenthal, Principal Investigator, and administered through Columbia University Medical Center. The opinions expressed in this paper are those of the authors and do not necessarily represent those of Merck Sharp & Dohme Corp.” 

We have removed this language from the manuscript and have included it within the cover letter. For the online submission form, the funding statement should read: “The study was supported in part by a research grant from Investigator-Initiated Studies Program of Merck Sharp & Dohme Corp grant awarded to Dr. Susan Rosenthal, Principal Investigator, and administered through Columbia University Irving Medical Center. The opinions expressed in this paper are those of the authors and do not necessarily represent those of Merck Sharp & Dohme Corp. The funders had no role in study design, data collection and analysis, decision to publish, or preparation of the manuscript.”

“Caroline Katzman, Julen Harris, Christine Mauro, Ariel de Roche, and Tucker Morgan have no conflicts of interest to declare. Gregory Zimet serves as an external advisory board member for Pfizer and Moderna, and as a consultant to Merck. He also has received investigator-initiated research funding from Merck administered through Indiana University and serves as an unpaid member of the Board of Directors for the Unity Consortium, a non-profit organization that promotes adolescent health through vaccination. Susan Rosenthal has received investigator-initiated research funding from Merck Investigator Studies Program administered through Columbia University Irving Medical Center.”

We have added this section, along with the statement above, to our Cover Letter. This section now reads: “Caroline Katzman, Julen Harris, Christine Mauro, Ariel de Roche, and Tucker Morgan have no conflicts of interest to declare. Gregory Zimet serves as an external advisory board member for Pfizer and Moderna, and as a consultant to Merck. He also has received investigator-initiated research funding from Merck administered through Indiana University and serves as an unpaid member of the Board of Directors for the Unity Consortium, a non-profit organization that promotes adolescent health through vaccination. Susan Rosenthal has received investigator-initiated research funding from Merck Investigator Studies Program administered through Columbia University Irving Medical Center. This does not alter our adherence to PLOS ONE policies on sharing data and materials.”

All data necessary to replicate this study’s findings will be made available without restriction at data repository ICPSR. 

7. Your ethics statement should only appear in the Methods section of your manuscript. If your ethics statement is written in any section besides the Methods, please move it to the Methods section and delete it from any other section. Please ensure that your ethics statement is included in your manuscript, as the ethics statement entered into the online submission form will not be published alongside your manuscript.

The Ethics Statement was moved to the Methods section of the paper. The Institutional Review Board of Columbia University Irving Medical Center approved this study as an exempt protocol on December 22, 2020. (IRB-AAAT5154).

8. Please include a caption for figure 1.

The caption has been added.

9. Please include your tables as part of your main manuscript and remove the individual files. Please note that supplementary tables (should remain/ be uploaded) as separate "supporting information" files

The tables have been included in the manuscript. The supplemental tables will be uploaded as supporting information. 

Review Comments to the Author:

Reviewer #1: This survey conducted in the United States adds to the evidence base concerning factors that drive one of the biggest global public health threats, i.e. vaccine hesitancy. Unfortunately, the study population was very small, resulting in highly imprecise estimates. Nonetheless, the study is interesting, and I especially enjoyed how the authors applied the “Diffusion of Innovation” model to how people behave with respect to COVID-19 vaccines.

That said, I believe the manuscript would benefit from revisions in order to more clearly and transparently describe various methodological aspects as well as the study findings. I hope my comments and suggestions below will help improve the paper.

ABSTRACT:

1. I strongly suggest briefly describing the study methods instead of jumping directly from rationale to results. In doing so, please make sure to clearly define the population of interest (adults? children? older adults? others?), indicate where the study was conducted, and describe participants’ selection methods (are these individuals meants to be representative of the general population?).

We have added language about the study methods and included the above. 

2. Please indicate the size the survey population and whether/how this changed from Wave 1 to Wave 2. It’s hard to interpret the study findings without any idea of how many people participated. Relatedly, I would suggest including 95% confidence intervals of prevalence proportions reported here.

We have added the population size in Waves 1 and 2 and have included 95% confidence intervals of prevalence proportions here. 

INTRODUCTION:

1. As I am sure you know, COVID-19 is the disease caused by SARS-CoV-2, so the expression “COVID-19 infection” is not entirely correct and should be replaced with “SARS-CoV-2 infection”. Please revise the terminology throughout the manuscript to refer to COVID-19 or SARS-CoV-2 as appropriate.

Thank you for this point – this has been addressed throughout the manuscript.

2. Lines 64-66: I understand this may be intuitive, but I would suggest citing one or more references to support this statement.

We have added citations to support this. 

3. Line 83: Consider adding the following citation along with references 2-5: Basta NE et al. Factors Associated With Willingness to Receive a COVID-19 Vaccine Among 23,819 Adults Aged 50 Years or Older: An Analysis of the Canadian Longitudinal Study on Aging. Am J Epidemiol. 2022 May 20;191(6):987-998. doi: 10.1093/aje/kwac029.

Thank you for sharing this resource! We have included it in the manuscript. 

4. Lines 92-94: I would suggest rephrasing this sentence for clarity.

This line has been rephrased. 

5. Lines 96-99: As also mentioned with respect to the abstract, I do think that the population and country of study should be clearly indicated. I encourage you to revise this last section of the introduction to incorporate this information. While some factors certainly play a role in favoring or hindering vaccine uptake across countries, other are more context specific. Given the peculiar social and political climate of the United States, I believe it is important to highlight the study context early on, as this also affects methodological choices as well as the interpretation of findings.

We have revised the introduction to include the country of study.

METHODS:

1. Consider slightly reorganizing the methods section into sub-sections with their own sub-headings to more systematically describe: the study design, cohort characteristics/sampling strategies, data collection methods, survey structure and relevant variables, and data analysis.

We have re-organized the Methods section into sub-sections for clarity.

2. I am not familiar with this Ipsos panel, and it is unclear to me whether the survey was paper-based, by phone or online. Or perhaps multiple options were offered to participants to accommodate their needs and preferences?

The survey is online only, we have included additional information about the Ipsos panel in the Methods section.

3. What is the size of the Ipsos survey panel?

The panel has 60,000 adult respondents, we have included this in the manuscript.

4. Considering that you leveraged on an existing survey panel, can you please clarify if the survey described in this article was developed ad hoc by the study authors or was part of a larger survey covering other health or social aspects?

The study authors developed the survey for this paper, we have included this in the manuscript. 

5. If I understand correctly, Ipsos participants were invited to participate to your Waves 1 and 2 in addition to other surveys they were already involved in. Did you sample a subgroup of Ipsos participants or invited all of them to complete the questionnaire?

We sampled a subgroup of participants, we have included this and the exact numbers of panelists to the methods section of the manuscript. 

6. For clarity and transparency, it would be helpful to see the list of questions, response options along with the resulting variables and their categorization. For example, you mention gender, race/ethnicity, political views, and other variables, but we don’t know how this was categorized. I would suggest including this information in a supplementary file. However, I do think that some key details should be explained in-text. For example, the definition of “general vaccine attitudes” should be more clearly described considering that the study is focused around this concept. So, knowing which six questions were used to calculate this score is extremely important to the reader. Consider summarizing this information in a table.

We have cited the group’s original paper which has a table with all questions from this survey and how the questions were scored. In addition, we have included examples of questions from each categories in-text for further clarity. Lastly, we have now included selected scaled questions from the Wave 1 Questionnaire used for this study in the Supporting Information files. 

7. Please clarify whether questionnaires administered at Wave 1 and Wave 2 include the exact same questions.

Wave 1 and 2 surveys were near-identical, however the Wave 2 survey was updated to reflect the changes in COVID-19 vaccine policy and protocol (i.e. the vaccine was available to all participants at the time of the Wave 2 survey so a question about eligibility was updated). This has now been stated explicitly in the Methods section of the manuscript.

8. Please explain how missing data were handled.

There were three sources of missing data in this study. The first was missing data due to people being loss-to-follow-up between Wave 1 and Wave 2. The Wave 2 weights provided by IPSOS and used in these analyses were adjusted to account for this.

The second source of missing data were participants who did not complete the main item on vaccination in wave 2 (n=12). Since this was such a small percentage of the overall sample (< 3%), they were simply removed from the analysis. The last source of missing data was on individual covariates of interest. These individuals were simply removed from the bivariate analyses as needed and never exceeded four observations lost.

We have included this in the results section.

9. How did you calculate weights?

Ipsos calculated the weights for the findings to be nationally representative, and these values were used for the analysis. We have added more detail about the Ipsos’ methodology for calculating weights in the Methods section. 

RESULTS:

1. As noted above, I think there is not enough clarity around the size of the study population. From Figure 1, we know that 1,189 people completed the Wave 1 survey and 843 completed the Wave 2 survey. It is still unclear what proportion of Ipsos participants were initially contacted and invited. Also, even if the numbers above are available in the flow diagram, I think it is important to include this information in-text as well, ideally in the Results’ opening sentence.

1991 participants were invited to “Wave 1” and 1062 participants were invited to “Wave 2,” this has been included in the first line of the Results section. 

2. I am quite surprised to see that only regression analysis results are presented in the article. Please include descriptive statistics (table and text) before heading to the results to the logistic regression model.

We have added a new table with descriptive statistics for the entire sample. We have also added a section in the Results that lays out the findings presented in this table. 

3. Lines 162-164: I find this sentence quite confusing: “Briefly, of the 509 participants who were vaccinated at Wave 1 and returned for follow-up, all but 3 reported being vaccinated by Wave 2 (99%).” Is these people were vaccinated at Wave 1, what were they supposed to report at Wave 2? Did you ask them if they had received a second dose? Or just to confirm their vaccination status as reported at Wave 1? Please rephrase and clarify.

We added language to state that this is likely due to participant error (these three participants reported vaccination initially and then stated they had not been vaccinated later). 

4. From the description of the results, I have trouble understanding whether there was any attempt to differentiate between first, second and booster doses. I believe this is an important point to discuss in the manuscript.

We added a sentence in the Methods section clarifying that we asked how many doses were received (only 2 doses were available at the time of the survey). However, for this analysis, we considered “vaccinated” as having received one vaccine dose. This is also discussed further in the limitations section of the paper. 

DISCUSSION:

1. I appreciate the discussion around the meaning and implications of the study findings. However, I strongly encourage you to revise it in order to incorporate considerations on whether and how your findings relate to other studies. What’s new in this study that was not captured in other studies? 

We have been more explicit in the Discussion about the ways in which our findings contribute to the literature, both in novel and in supportive ways. 

Do you think that your findings apply to other countries and why or why not? Also, what do your results tell us about attitudes towards vaccination in different parts of the US? 

Our data likely does not apply to other countries or various regions of the U.S. based on specific social/political contexts and vaccine availability. Additionally, this study was designed to be nationally representative but not regionally representative and thus cannot answer the question of attitudes toward vaccination in different parts of the US. We have added this to the “Limitations” section of the discussion. 

I know the size of the study population was quite small and you certainly lacked power to conduct any subgroup analyses, but – in my opinion – this is an important point to discuss.

Thank you for this point – we have added this to the limitations section. 

Reviewer #2: GENERAL COMMENTS TO AUTHORS

Overall this is a relatively well-designed and implemented study involving two KnowledgePanel surveys regarding COVID-19 vaccination intent and receipt- one fielded in April 2021 (when only some Americans were recommended to be vaccinated n the staged-in process due to vaccine supply) and the follow-up survey in February 2022 (when all Americans were vaccine-eligible). The study goal was to identify predictors of ultimate vaccination. The authors used a reasonable Diffusion of Innovation (DOI) conceptual model to ground their study and analysis. They found, unsurprisingly, that those who wanted the vaccine “ASAP” nearly all received it later, those who were cleary non-acceptors initially remained so afterwards, and those in between tended to be vaccinated with higher-income and barriers being moderate predictors in the wait-and-see group while belief structures were paramount in the non-acceptor group.

The study was done well, and the findings are believable. Key limitations are that Wave 1 occurred when some could get vaccinated so even the initial group was unusual (for example leaving out healthcare, essential workers, and older adults) from eligibility. Thus even prediction suffers from an unusual initial group. Second, sample sizes per DOI group were small (between 100-200 mostly), limiting confidence in prediction. Third, with the rapidly shifting background and concerns having to do with receipt of bivalent vaccine in a population which has largely already had COVID and largely already had at least 1 vaccination, the study implications at this point are limited.

SPECIFIC COMMENTS

Introduction

While well written, this section would be enhanced by relating the DOI and concepts to our current situation regarding COVID-19 vaccination and the likely future situation of having annual or biannual vaccine recommendations.

This is a great point. The importance of this data for future recommendations about bivalent boosters and annual/biannual vaccination was included to the introduction. 

Methods

In general the methods were described well.

The authors could include a bit more on the underlying KnowledgePanel methods (recognizing that they published results of their original wave 1 study.

More information about the KnowledgePanel was included in this draft of the manuscript.

Results

The findings are presented relatively well and clearly

Thank you for this feedback!

Discussion

This section is well thought out. It could be improved by increasing the discussion of study findings as it relates to the current situation of the US recommending bivalent vaccination for a population that has largely had COVID infection and has had at least one prior vaccination.

Thank you for this point – we have added a section about the evolving landscape of COVID-19 vaccination and the continued importance of these findings.

The discussion about relationship to influenza vaccination is not new and does not currently lend itself to actions. The authors should consider discussing how knowledge of influenza vaccination might practically help providers or health systems in their attempt to promote COVID-19 vaccination (beyond just showing the association). In addition, future COVID-19 vaccines might be combined with influenza vaccines so a discussion could cover that topic.

Thank you for this – we have added this important point to the discussion.

---

## [Decision Letter · Decision Letter 1]

21 Jul 2023

Longitudinal assessment of COVID-19 vaccine uptake: A two-wave survey of a nationally representative U.S. sample

PONE-D-23-06419R1

Dear Dr. Katzman,

We’re pleased to inform you that your manuscript has been judged scientifically suitable for publication and will be formally accepted for publication once it meets all outstanding technical requirements.

Kind regards,

Jerome Nyhalah Dinga, PhD

Academic Editor

PLOS

---

## [Editor Report · Acceptance letter]

26 Sep 2023

PONE-D-23-06419R1 

Longitudinal assessment of COVID-19 vaccine uptake: A two-wave survey of a nationally representative U.S. sample 

Dear Dr. Katzman:

I'm pleased to inform you that your manuscript has been deemed suitable for publication in PLOS ONE. Congratulations! Your manuscript is now with our production department. 

Kind regards, 

on behalf of

Dr. Jerome Nyhalah Dinga 

Academic Editor

PLOS ONE